# The impact of bilateral ongoing activity on evoked responses in mouse cortex

**Daisuke Shimaoka[1†], Nicholas A Steinmetz[1,2‡], Kenneth D Harris[2], Matteo Carandini[1]\***

[1]UCL Institute of Ophthalmology, University College London, London, United Kingdom; [2]UCL Institute of Neurology, University College London, London, United Kingdom

**Abstract** In the absence of external stimuli or overt behavior, the activity of the left and right cortical hemispheres shows fluctuations that are largely bilateral. Here, we show that these fluctuations are largely responsible for the variability observed in cortical responses to sensory stimuli. Using widefield imaging of voltage and calcium signals, we measured activity in the cortex of mice performing a visual detection task. Bilateral fluctuations invested all areas, particularly those closest to the midline. Activity was less bilateral in the monocular region of primary visual cortex and, especially during task engagement, in secondary motor cortex. Ongoing bilateral fluctuations dominated unilateral visual responses, and interacted additively with them, explaining much of the variance in trial-by-trial activity. Even though these fluctuations occurred in regions necessary for the task, they did not affect detection behavior. We conclude that bilateral ongoing activity continues during visual stimulation and has a powerful additive impact on visual responses.
DOI: https://doi.org/10.7554/eLife.43533.001

**\*For correspondence:**
m.carandini@ucl.ac.uk

**Present address:** [†]Biomedicine Discovery Institute & Department of Physiology, Monash University, Clayton, Australia; [‡]Department of Biological Structure, University of Washington, Seattle, United States

**Competing interests:** The authors declare that no competing interests exist.

## Introduction

Even in the absence of external stimuli or overt behavior, the patterns of activity exhibited by the awake cerebral cortex, known as ongoing activity, can display a striking degree of bilateral symmetry. When one region of cortex becomes active, this activity is often accompanied by activity in the symmetrical region in the opposite hemisphere. This bilateral symmetry has been documented both in awake mice (*Mohajerani et al., 2010*) and in awake humans (*Fox et al., 2006*; *Tyszka et al., 2011*). The degree of symmetry varies across cortical areas (*Mohajerani et al., 2010*), but the rules behind the differences across areas remain unknown.

Bilateral ongoing activity coexists with event-related activity, which is not necessarily bilateral. For instance, when human subjects move their right hand, the movement-related BOLD (Blood-oxygen-level-dependent) response in left motor cortex coexists with bilateral ongoing fluctuations in both left and right motor cortices (*Fox et al., 2006*).

It is not clear, however, to what extent this bilateral ongoing activity can influence sensory-evoked activity. Similar to the activity associated with movements, the activity evoked by visual stimuli in cortex is not generally bilateral. For instance, a visual stimulus presented to the side of the animal would evoke unilateral responses in the contralateral visual cortex. It is not known whether the bilateral ongoing activity coexists with the visually evoked activity, or whether it is quenched by the onset of visual responses (*Churchland et al., 2010*; *He, 2013*).

If bilateral ongoing activity persists during sensory responses, it could contribute to the large trial-by-trial variability seen in these responses. Even in response to an identical visual stimulus, neural responses in primary visual cortex are well-known to vary strongly from trial to trial (*Tolhurst et al., 1983*). This variability has been observed not only in the spike trains but also in intracellular recordings from single neurons (*Anderson et al., 2000*; *Carandini, 2004*) and in voltage

imaging of neural populations (*Arieli et al., 1996*; *Carandini et al., 2015*). How much of this trial-by-trial variability is explained by bilateral ongoing activity?

If much of trial-by-trial variability is explained by bilateral ongoing activity, then bilateral measurements may become necessary to estimate true sensory responses. How would such a procedure compare to the standard method (*Arieli et al., 1996*) of measuring activity before a stimulus and removing it from the subsequent activity? Moreover, it is not known how bilateral ongoing activity interacts with visual responses, for example whether the interaction can be described as simply additive or multiplicative (*Arieli et al., 1996*; *Ecker et al., 2014*; *Goris et al., 2014*; *He, 2013*; *Lin et al., 2015*; *Schölvinck et al., 2015*). Likewise, it is not known whether the bilateral fluctuations impair stimulus coding (*Averbeck et al., 2006*) and therefore impact perception.

To address these questions, we performed widefield optical imaging from both cortical hemispheres of mice performing a visual detection task. Widefield imaging allowed us to monitor bilateral ongoing activity and visually evoked response across hemispheres. To observe these forms of activity both in membrane potential and in spiking activity, we measured both from mice expressing a voltage sensor and from mice expressing a calcium sensor. These measurements revealed that bilateral ongoing activity has a prominent impact on visually evoked responses. An additive model for the interaction of ongoing and evoked activities provided excellent fits to the data, explaining up to 60% of trial-by-trial variance in the observed imaging signal. On the other hand, bilateral ongoing activity did not seem to affect perceptual decisions, suggesting that it does not impact stimulus coding.

## Results

We performed widefield imaging in the cerebral cortex of transgenic mice expressing voltage indicator VSFPB1.2 or calcium indicator GCaMP6 in excitatory neurons (*Figure 1*). The imaging window covered the dorsal part of both cortical hemispheres. We defined the location of sensory and motor areas by aligning the images to the Common Coordinate Framework of the Allen Brain Atlas (*Figure 1A*). We confirmed the outlines of the visual areas based on functional mapping (visual field sign, *Figure 1B*).

We then measured activity while the mice performed a visual two-alternative task (*Burgess et al., 2017*; *Zatka-Haas et al., 2019*). In the task, mice reported the location of a grating stimulus presented on the left or right visual field. Before the stimulus application, the mice were required to hold the wheel still for at least 500 ms. We begin by examining the ongoing activity measured in this pre-stimulus interval, focusing on trials when mice held the wheel without making eye movements.

### Ongoing cortical activity is largely bilateral

During the pre-stimulus interval, we observed strong bilateral voltage activity over the entire dorsal cortex (*Figure 1C–D*). This activity involved multiple cortical areas (*Figure 1C*) and was not accompanied by any obvious body movement (*Figure 1—video 1*). As reported in previous studies (*Mohajerani et al., 2010*; *Shimaoka et al., 2017*), activity was largely symmetric between the left and right hemispheres. Indeed, this symmetric, bilateral activity accounted for most of the power of the measured voltage signals, at all measured frequencies (*Figure 1D*): we did not observe significant difference in coherence across frequencies (2–7 Hz, p>0.05, one-way ANOVA).

These phenomena were not limited to voltage fluctuations, which might in principle be entirely subthreshold: they were also seen in calcium signals, which reflect neuronal spiking (*Figure 1E–F*). Calcium activity during the pre-stimulus interval involved multiple cortical areas and was also typically symmetric between the hemispheres (*Figure 1E*, *Figure 1—video 1*). Just as with voltage signals, bilateral calcium activity accounted for most of the power of the measured calcium signals, and seemed to do so at all measured frequencies (*Figure 1F*).

### Bilateral ongoing fluctuations differ across areas

The bilateral symmetry of ongoing activity, however, was not equal for all cortical areas, and tended to decrease with distance from the midline (*Figure 2*). For each cortical area of interest, we computed the temporal correlation of that area ('seed area') against all the other pixels. Both in mice-expressing voltage indicators (*Figure 2A*) and in mice-expressing calcium indicators (*Figure 2B*), the resulting seed-area maps showed high correlation in the corresponding region in the opposite

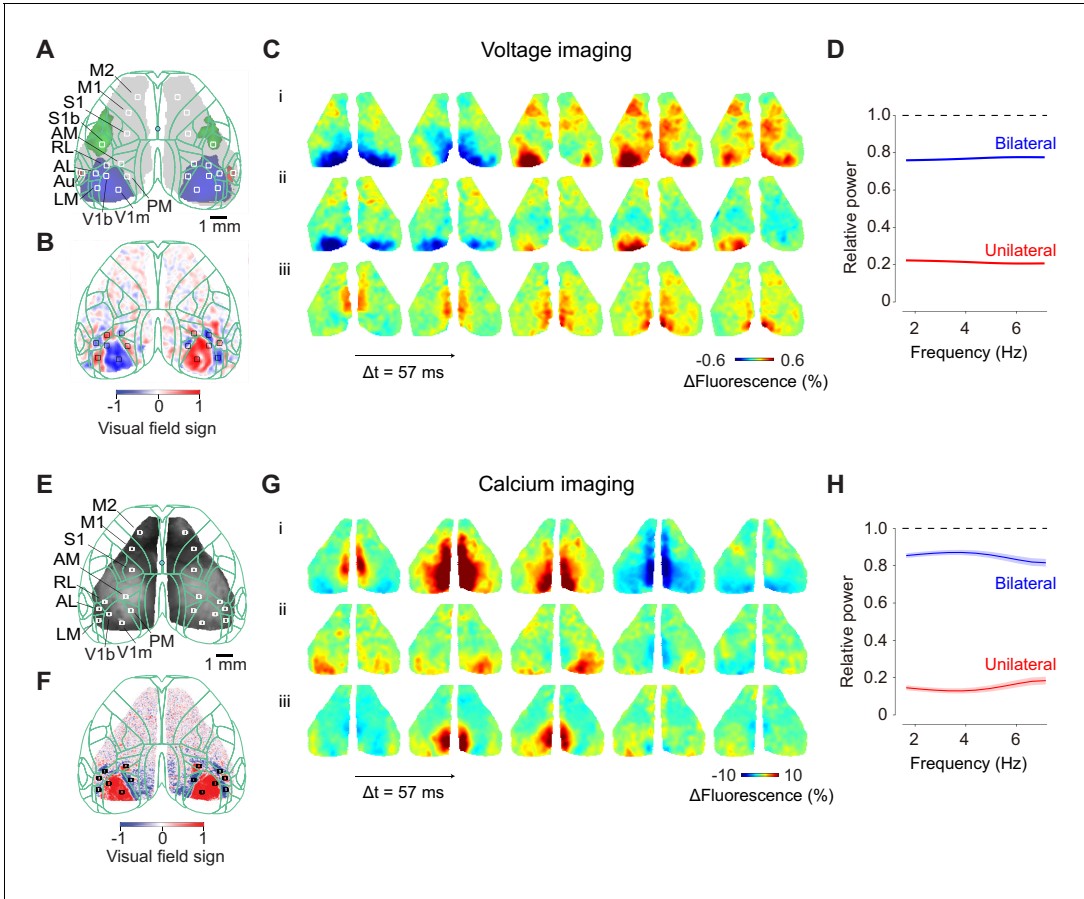

**Figure 1.** Ongoing cortical activity is largely bilateral. (**A**) Location of somatosensory and motor areas based on the anatomical database by Allen Brain Atlas. The background image indicate a raw fluorescence signal obtained from voltage imaging. Superimposed green contour lines indicate borders registered in the Common Coordinate Framework of the Allen Brain Atlas. (**B**) Location of individual visual areas, identified by visual field sign. The border of its sign indicates the border of visual areas. (**C**) Three example sequences of voltage signals during the period before visual stimulation when a mouse was holding the wheel without detectable eye movements. (**D**) Relative power spectral density of the bilateral and unilateral ongoing voltage components with respect to measured ongoing voltage signal, averaged across the visual areas from the example mouse. Different color indicates symmetrical component of activities between the hemispheres, computed as (left + right)/2 (*blue*), asymmetrical component of activities, computed as (left-right)/2 (*red*). Both components were divided by the total signal measured in each frequency. Background shades indicate S.E. across two animals. (**E**: Same as **C**), for widefield imaging of calcium signals. (**F**: Same as **D**), for seven mice expressing calcium indicators. The video of activity in these three trials is available at *Figure 1—video 1*.

DOI: https://doi.org/10.7554/eLife.43533.002

The following video and figure supplement are available for figure 1:

**Figure supplement 1.** Pre-processing of the imaging signals has no effect on bilaterality.

DOI: https://doi.org/10.7554/eLife.43533.003

**Figure 1—video 1.** Ongoing cortical activity is largely bilateral.

DOI: https://doi.org/10.7554/eLife.43533.004

hemisphere. This bilateral correlation was particularly strong in primary somatosensory cortex (S1) and in medial visual areas such as PM (*Figure 2C,D*). Bilateral correlation tended to decrease with a region's distance from the midline, both in mice-expressing voltage indicators (*Figure 2C*, pairwise correlation ρ = −0.44) and in mice-expressing calcium indicators (*Figure 2D*, pairwise correlation ρ = −0.42).

However, correlation was lower than expected in the monocular region of primary visual cortex, V1m and, when the animal was engaged in the task, in secondary motor cortex (M2, *Figure 2C,D*). In area M2, this low bilateral correlation may be related to task engagement, because bilateral correlation seemed to be higher when the animal was in a passive condition (*Figure 2C,D*, white circle). Indeed, the reduction in bilateral fluctuations seen with task engagement was specific to area M2,

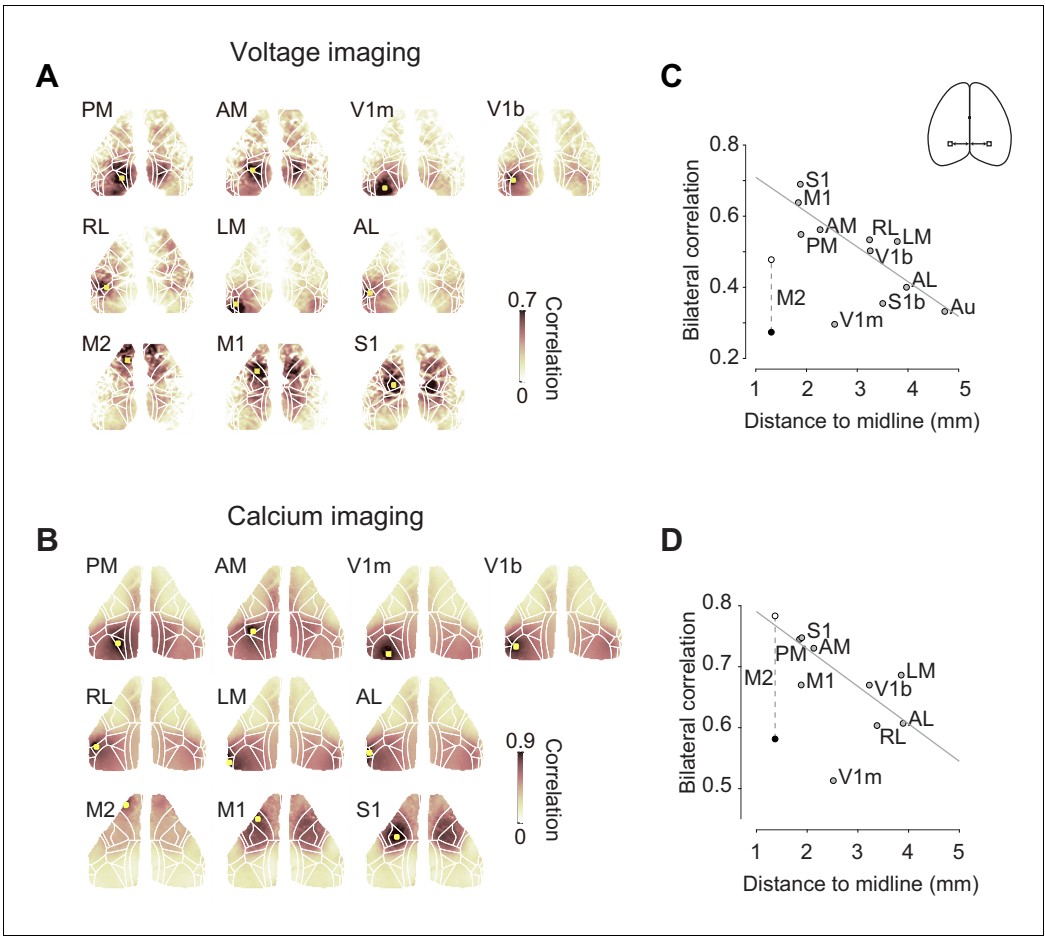

**Figure 2.** Bilateral fluctuations differ across areas. (**A**) Seed-area maps of correlation from an example animal expressing a voltage indicator. Correlation was computed against the average signal within the red square to signals of the all the other pixels. (**B**, as in **A**), for an animal expressing a calcium indicator. (**C**) Correlation between left and right symmetrical ROIs (Regions of interest), as a function of distance ROI and the midline. Each dot represents average of two mice expressing voltage indicators. Data for area M2 is shown during passive (*white*) and active (*black*) conditions. Data for other areas are averages of active and passive conditions (*gray*). Gray line indicates a result of linear fit without using M2 and V1m. (**D**) Same as (**C**), for four mice expressing a calcium indicator.

DOI: https://doi.org/10.7554/eLife.43533.005

The following figure supplement is available for figure 2:

**Figure supplement 1.** Bilateral correlation during the task and during the passive condition.
DOI: https://doi.org/10.7554/eLife.43533.006

where it was significant (*Figure 2—figure supplement 1*). For each cortical area, we compared bilateral correlation in the active task vs. the passive condition using VSFP and GCaMP data. For all areas except M2, there was no significant change in correlations across the two conditions. This included V1m, where correlations were lower than in other areas but not significantly different across the two conditions ($p > 0.05$ after Bonferroni correction, paired *t* test, n = 6 mice, *Figure 2—figure supplement 1D*). In M2, instead, there was a significant reduction during the active task ($p = 0.0013/0.013$ before/after Bonferroni correction, paired *t* test, n = 6 mice, *Figure 2—figure supplement 1H*).

## Bilateral ongoing fluctuations continue during unilateral visual responses

Bilateral ongoing fluctuations continued following the onset of the visual stimulus and were typically larger than the unilateral visual responses (*Figure 3*). For example, consider the voltage responses

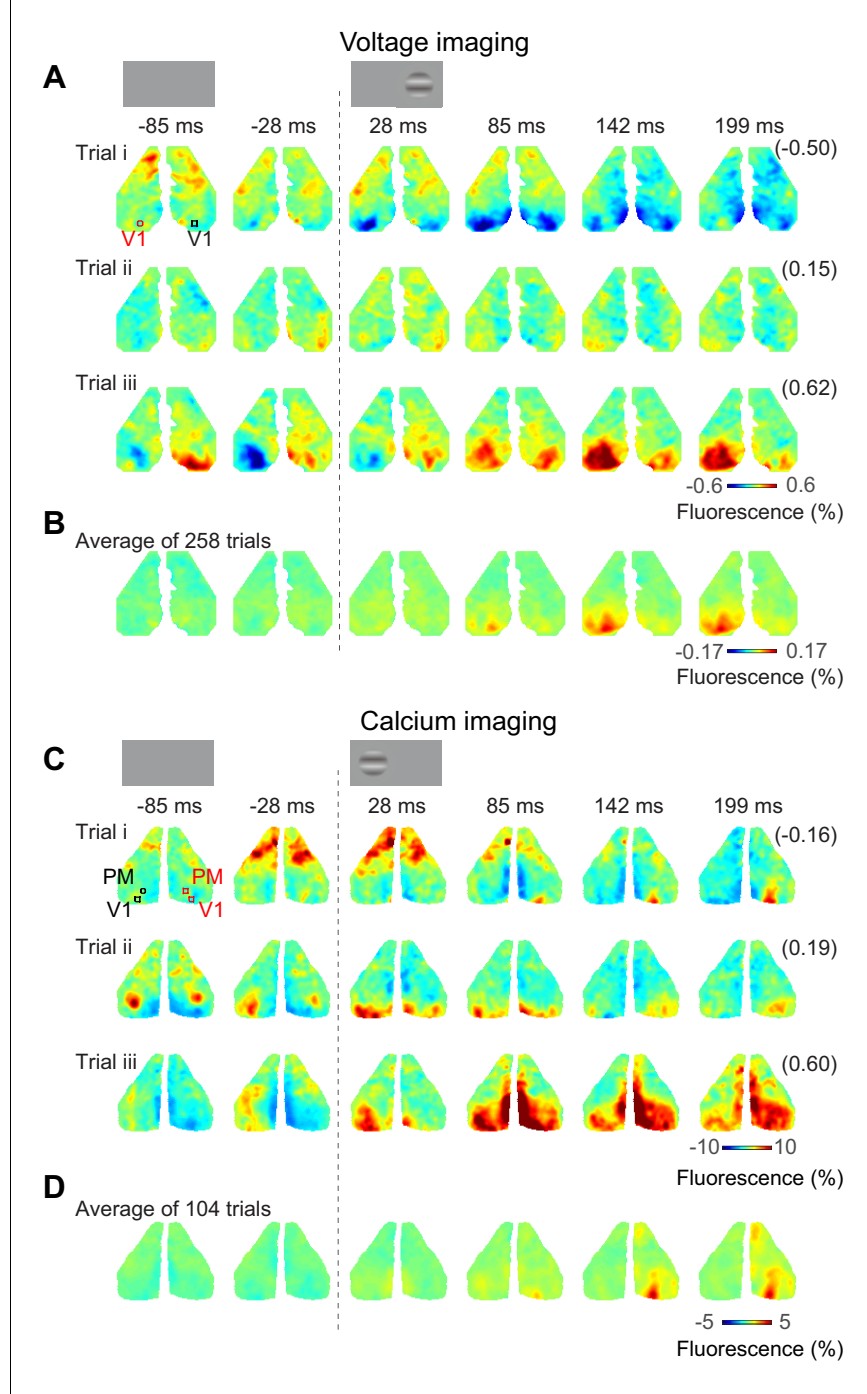

**Figure 3.** Bilateral ongoing activity continues during unilateral visual responses. (**A**) Three example sequences of voltage signal during the period after presentation of 50% contrast to the left monocular visual field in an example mouse expressing VSFP. In each examples, the right-most number in a parenthesis indicates correlation coefficient of the sequence to the average across repeats (**B**). (**B**) Average neural activity across repeats. (**C**, **D**) Same as (**A**), (**B**), for a mouse expressing calcium indicator. The video of activity in these three trials is available at *Figure 3—video 1*.

DOI: https://doi.org/10.7554/eLife.43533.007

The following video is available for figure 3:

**Figure 3—video 1.** Bilateral ongoing activity continues during unilateral visual responses.

DOI: https://doi.org/10.7554/eLife.43533.008

to three presentations of a 50% contrast stimulus in the right monocular visual field (*Figure 3A*). In all these trials, activity before visual stimulation was largely bilateral and continued to fluctuate after stimulus onset. Only by acquiring multiple trials could these ongoing fluctuations be averaged out to reveal the underlying unilateral response to the visual stimulus (258 trials, *Figure 3B*). Similar results were obtained with calcium imaging. The bilateral activity seemed to continue unperturbed following presentation of a stimulus in the left monocular visual field (*Figure 3C*, *Figure 3—video 1*). Averaging the calcium activity over 104 trials revealed localized visual responses in area V1 and higher visual areas, which grew between 50 and 150 ms after stimulus onset (*Figure 3D*).

Confirming these observations, we measured the activity evoked by our stimuli in areas V1 and PM, and observed that in the first 150 ms following stimulus presentation this activity was exclusively unilateral (*Figure 4*). Task stimuli elicited visual activity solely in contralateral cortex, both in area V1 (*Figure 4A*, *red traces*) and in area PM (*Figure 4B*, *red traces*). The mean activity rose in the period 50–150 ms after stimulus onset, and increased with contrast as expected (*Figure 4E,F*). Consistent with known variability of neural activity, this increase in mean was accompanied by an increase in standard deviation across trials (*Figure 4C,D*). By contrast, in the unstimulated hemisphere, the stimuli affected neither the mean activity (*Figure 4A,B*, *black traces*) nor the standard deviation of activity (*Figure 4C,D*, *black traces*). These results suggest that bilateral fluctuations are not reduced by the onset of visual stimuli.

## Bilateral fluctuations and visually evoked responses add linearly

We exploited the unilateral nature of visual responses to study their interaction with bilateral fluctuations. We illustrate these analyses with calcium imaging, where we have larger numbers of animals.

Inspection of trial-by-trial calcium signals suggested an additive interaction of stimulus-evoked responses and bilateral ongoing activity (*Figure 5A,B*). Consider, for instance, the trial-by-trial fluorescence measured in left and right area PM following the onset of a 100% contrast stimulus on the left (*Figure 5A,B*). Activities were highly variable across trials both in the left, unstimulated

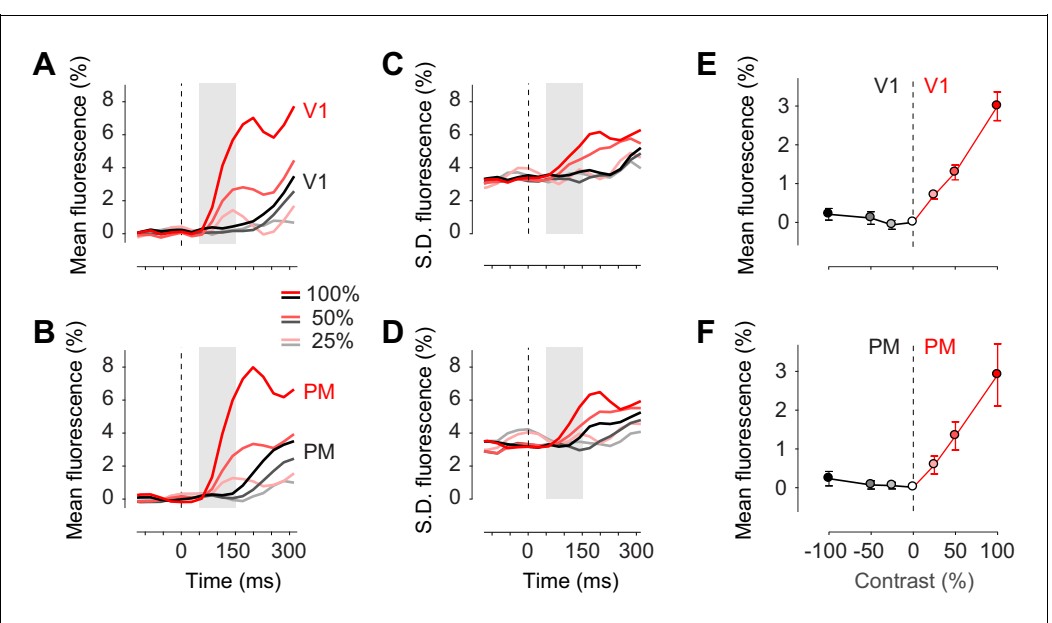

**Figure 4.** Task stimuli evoke unilateral responses and do not quench bilateral fluctuations. (A) Mean calcium signal across trials from seven animals in the right stimulated V1 (*red*) and in the left unstimulated V1 (*black*). Time period of 50–150 ms after stimulus onset (*shaded area*) was used for subsequent analysis of visually evoked response. (B) Same as (A) in PM. (C.D) Same as panels (A) and (B) for standard deviation across trials. (E) Average neural activity 50–150 ms after visual stimulation to different contrasts (contrast-response curve) in V1. Activity in the stimulated and unstimulated V1 is shown as red and black lines. Error bars indicate S.E. across seven animals. (F) Same as panel (E) in PM.
DOI: https://doi.org/10.7554/eLife.43533.009

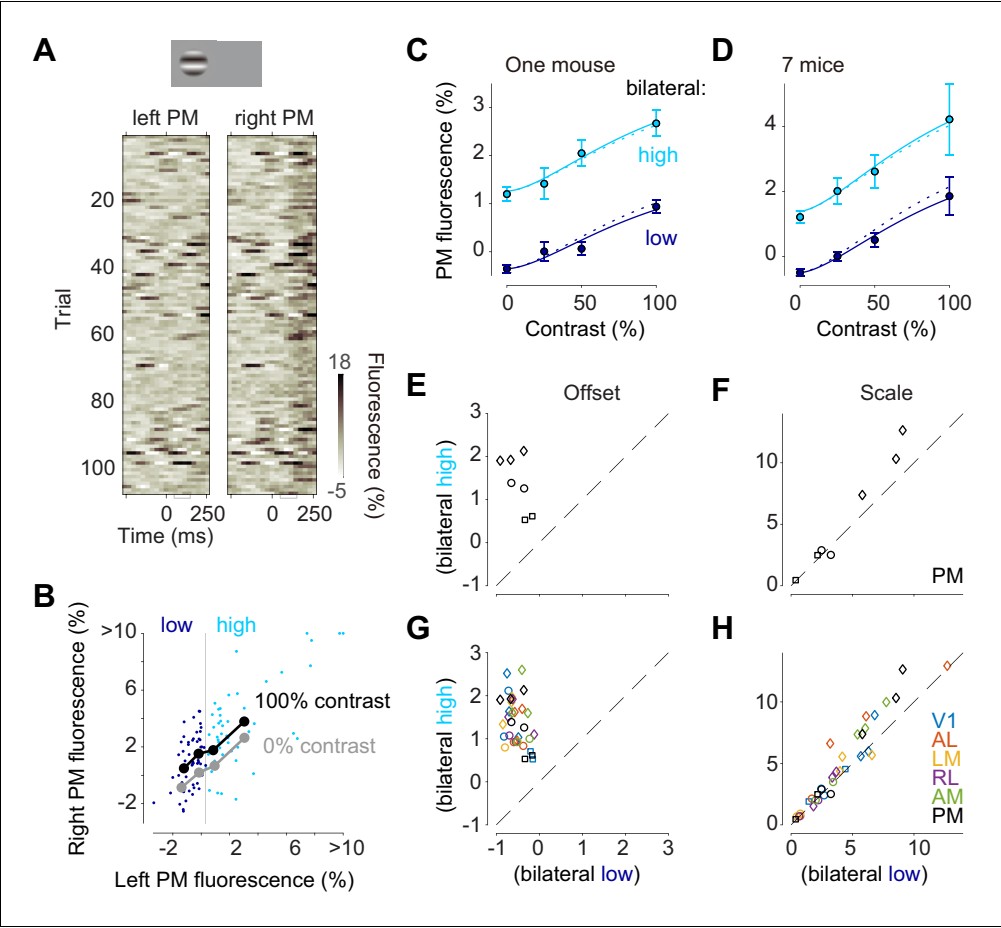

**Figure 5.** Bilateral ongoing activity and visually evoked responses add linearly. (A) Single trial traces aligned by the onset of visual stimulation of 100% contrast to the left visual field in the left unstimulated PM and right stimulated PM. The time period (50–150 ms) used for the analysis of visually evoked response is indicated as a gray bracket. (B) Relationship between left PM (*abscissa*) and right PM (*ordinate*) 50–150 ms after visual stimulation. Single trial activity to 100% contrast stimulation is shown as cyan and blue dots. Moving average along left PM is shown as thick black (100% contrast) and gray (0% contrast) lines. (C) Contrast responses from trial groups of high (*cyan*) and low (*blue*) contralateral ongoing activity in one animal. Data points and error bars indicate median and median absolute deviation across trials from one animal. Curves indicate contrast-response functions where scale parameter is free (*solid curve*) or fixed (*dotted curve*) between the high and low ongoing activities. (D) Contrast responses, average of seven animals. Error bars represent S.E. across animals. (E, F) Parameters of contrast-response functions across animals. Offset (E) and scale (F) parameters were estimated individually from trial groups of low (*abscissa*) and high (*ordinate*) ongoing activities. Genetic background was Snap25-GCaMP6s (*circles*), TetO-GCaMP6s;Camk2a-tTA (*diamonds*) or Vglut1-Cre;Ai95 (*squares*). (G, H) Same as panels (E) and (F), for all the visual areas.

DOI: https://doi.org/10.7554/eLife.43533.010

The following figure supplements are available for figure 5:

**Figure supplement 1.** Bilateral ongoing activity and visual responses interact additively regardless of the similarity of their patterns.
DOI: https://doi.org/10.7554/eLife.43533.011

**Figure supplement 2.** The effects of bilateral ongoing activity were due to fast changes in cortical activation, not to slow changes in cortical state.
DOI: https://doi.org/10.7554/eLife.43533.012

**Figure supplement 3.** The effects of bilateral ongoing activity were due to fast changes in cortical activation, not to slow changes in pupil diameter.
DOI: https://doi.org/10.7554/eLife.43533.013

**Figure supplement 4.** Ongoing activity sampled as prestimulus activity.

*Figure 5 continued on next page*

*Figure 5 continued*

DOI: https://doi.org/10.7554/eLife.43533.014

hemisphere and in the right, stimulated hemisphere (*Figure 5A*). This variability persisted at the time of the visual response, 50–150 ms after stimulus onset (*Figure 5A*), and showed strong bilateral correlation (*Figure 5B*, *dots*).

To quantify the relationship between left and right PM in the response period we computed a sliding average of right PM activity as a function of left PM activity, separately for 100% contrast visual stimulus (*Figure 5B black curve*), and 0% contrast stimulus (*Figure 5B gray curve*). Consistent with an additive interaction, the effect of the stimulus was to elevate this right-hemisphere activity by a constant amount, irrespective of left-hemisphere activity (*Figure 5B*, *black* vs. *gray curves*).

To test whether the interaction between ongoing activity and visually evoked responses is additive, we examined the effects of variations in each factor's amplitude: the contrast of the stimulus, and the simultaneous amplitude of the bilateral fluctuation (*Figure 5C–H*). We divided trials into two groups according to fluorescence at the time of the responses (50–150 ms) in the unstimulated hemisphere (*Figure 5B*; *blue* vs. *cyan*). For all stimulus contrasts, the visual responses obtained when ongoing activity was high (*Figure 5C,D*, *cyan*) appeared additively shifted compared to those obtained when ongoing activity was low (*Figure 5C,D*, *blue*). To quantify this effect, we fitted the contrast responses with a curve with two free parameters: offset and scale. The offset parameter was significantly higher when ongoing activity was high ($p = 0.0009$, paired $t$ test, $n = 7$ mice *Figure 5E*). The scale parameter was much less affected: it appeared to show a mild increase when bilateral activity was high, but this increase was not significant ($p > 0.05$, paired $t$ test, $n = 7$ mice *Figure 5F*). This result held not only for PM but also for all other visual areas: bilateral ongoing activity changed offset but not scale ($n = 7$ in V1, $n = 5$ mice in AL, LM and RL, *Figure 5G,H*). Similar results were seen when we chose different criteria for distinguishing high vs. low ongoing activity.

The additive interaction between bilateral ongoing activity and visual responses held regardless of the degree to which ongoing activity resembled visual responses (*Figure 5—figure supplement 1*). Ongoing activity sometimes resembles the responses evoked by visual stimulation (*Kenet et al., 2003*; *Mohajerani et al., 2013*). To assess whether this similarity affects the interaction between ongoing activity and visually evoked responses, we created a template based on the average response to a visual stimulus, made it bilateral, and divided trials into two groups according to correlation with this template at the time of the stimulus onset. For all stimulus contrasts, the visual responses obtained when ongoing activity was positively or negatively correlated with the template differed by a single additive factor (Offset: $p > 0.05$ in V1 $n = 7$ mice, $p < 0.05$ in other visual areas $n = 5$ mice, paired $t$ test; Scale: $p > 0.05$ in all areas, paired $t$ test, $n = 7$ mice in V1, $n = 5$ mice in AL, LM, and RL, *Figure 5—figure supplement 1F,G*).

Bilateral fluctuations are rapid and their effect is independent of arousal. These effects of bilateral ongoing activity were due to fast changes in cortical activation, not to slow changes in cortical state (*Figure 5—figure supplement 2*). Slow state changes might in principle have been able to explain these results; for instance, there may be more opportunities for bilateral ongoing activity to be 'high' when the cortex is in a more synchronized state, and this state might be the fundamental factor that leads to elevated visual responses. To assess slow changes in cortical state, we computed the amplitude of 2–7 Hz fluctuations in bilateral activity before the stimulus, and thus divided trials in four groups, for all combinations of state (more desynchronized vs. more synchronized) and instantaneous activity (low vs. high, *Figure 5—figure supplement 2A*). The interaction between instantaneous ongoing activity and visually evoked responses was seen in synchronized and desynchronized states (*Figure 5—figure supplement 2B,C*). The offset associated with high bilateral activity seemed to be larger in the synchronized state (*Figure 5—figure supplement 2C*). However, this larger offset was explained by the fact that in the synchronized state, the bilateral ongoing activity tended to reach higher levels (*Figure 5—figure supplement 2A,D*). Similar results were seen across the other mice (*Figure 5—figure supplement 2E–G*): fast fluctuations, rather than cortical state, determine the size of the additive interaction on visually evoked responses.

Similarly, the effects of bilateral ongoing activity were not related to slow changes in level of arousal (*Figure 5—figure supplement 3*). We gauged the level of arousal by measuring pupil

diameter, and we divided the trials according to pupil diameter (small vs. large, *Figure 5—figure supplement 3A*), and found that the effect of bilateral activity was present in both cases (*Figure 5—figure supplement 3B,C,E,F*). Once again, responses could be explained based on fast ongoing fluctuations, with no need to include pupil dilation as a predictor (*Figure 5—figure supplement 3D, G*).

Because ongoing activity changed rapidly and was largely bilateral, its impact could not be estimated through the typical procedure of measuring unilateral activity before stimulus onset (*Figure 5—figure supplement 4*). Ongoing activity is often estimated from prestimulus measurements (*Arieli et al., 1996*), and these estimates are often subtracted from the subsequent measurements. This procedure is likely to be less effective in the face of strong, fast bilateral ongoing activity. Indeed, the cross-correlation between corresponding regions on the two hemispheres typically decreased to 0 in ~100 ms, and in some mice became negative afterwards (*Figure 5—figure supplement 4A*). As a result, estimates of bilateral ongoing activity obtained from the unstimulated hemisphere were superior to estimates obtained in the stimulated hemisphere before stimulus onset. This can be seen by dividing all trials into four groups, for all combinations of prestimulus activity in the stimulated hemisphere (down vs. up) and poststimulus activity in the unstimulated hemisphere (low vs. high *Figure 5—figure supplement 4B*). The strong effect of bilateral activity was present in both up and down groups (*Figure 5—figure supplement 4C,D,F,G*), and responses could be explained based on fast bilateral fluctuations (*Figure 5—figure supplement 4E,H*).

## Bilateral fluctuations do not affect perceptual reports

Do these fast fluctuations in bilateral ongoing activity result in percepts? On the one hand, one might imagine that they do, because they impact cortical areas that are necessary for the performance of the task (*Burgess et al., 2017*; *Zatka-Haas et al., 2019*), so one might expect that whatever affects their activity will influence the percept of the animal. On the other hand, as the correlation structure of bilateral spontaneous activity differs from that of unilateral sensory responses, it may have little impact on information coding (*Averbeck et al., 2006*): interpreted naively, bilateral activations would constitute stimuli with a peculiar mirror symmetry across the vertical midline, which would be unlikely to be confused with actual images.

Bilateral ongoing fluctuations did not affect the behavior of the mouse in the task (*Figure 6*). We divided trials according to whether bilateral ongoing activity in PM was low or high (as in *Figure 5C, D*), and computed psychometric curves in each group. The two psychometric curves were essentially identical: there was no effect of bilateral activity on the probability of choosing the correct side (*Figure 6A,C*), or to choose a no-go response, which would be appropriate when stimulus is zero (*Figure 6B,D*). We fitted the behavioral responses with a simple probabilistic observer model (*Burgess et al., 2017*; *Zatka-Haas et al., 2019*), and measured offset and scale parameters of the perceptual contrast sensitivity function inferred by the model. Neither offset nor scale were significantly affected by bilateral ongoing activity (p>0.05, paired $t$ test, n = 7 mice, *Figure 6E,G*). Similar results were seen when we used other visual areas to measure bilateral activity (p>0.05, n = 7 mice in V1, n = 5 mice in AL, LM, and RL, *Figure 6F,H*).

These results indicate that the bilateral ongoing activity did not affect perceptual reports. We next asked whether this result could be explained by the bilateral activity having very different features from the visually evoked activity. To test this hypothesis, we restricted the analysis to cases when the ongoing activity resembled the visual responses (*Mohajerani et al., 2013*). Even in these cases, we saw no significant effects on the performance of the animal (p>0.05, n = 7 mice, *Figure 6—figure supplement 1*).

## Bilateral fluctuations dominate trial-by-trial variations in evoked visual responses

To compare the effects of bilateral ongoing fluctuations and evoked visual responses we fit a simple additive model (*Figure 7A*). In the model, the activity in a predicted visual area in trial $i$ at time $t$ from stimulus onset is the sum of a visual component $V(c_i, t)$, which depends only on contrast $c_i$, and a contribution from bilateral fluctuation $F_i(t)$, that varies across trials independently of the stimulus

$$f_i(t) = V(c_i, t) + F_i(t). \tag{1}$$

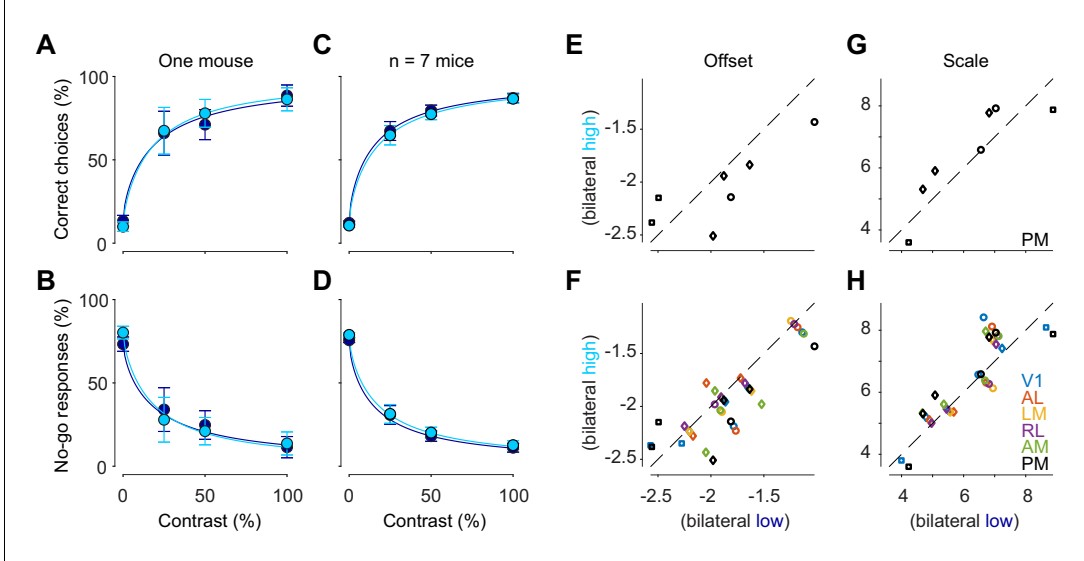

**Figure 6.** Bilateral ongoing activity does not affect perceptual reports. (**A**) Probability of turning the wheel to the correct direction as a function of stimulus contrast in one animal. The probability is computed separately in trial groups of low (*blue*) and high (*cyan*) ongoing activity measured in area PM. Error bars indicate 95% binomial confidence intervals. Curves indicate psychometric functions obtained by fitting a probabilistic observer model (**Burgess et al., 2017**) separately to the two sets of responses. (**B**) Same as (**A**), for probability of no-go. (**C, D**) Same as (**A**), (**B**), averaged across mice. Error bars represent S.E. across seven mice. (**E**) Offset parameter of the perceptual contrast sensitivity function inferred by the model, measured when bilateral activity in PM was low (*abscissa*) vs. high (*ordinate*). Symbols indicate parameters from individual animals. Genetic background was Snap25-GCaMP6s (*circles*), TetO-GCaMP6s;Camk2a-tTA (*diamonds*) or Vglut1-Cre;Ai95 (*squares*). (**F** Same as **E**), for all visual areas. (**G, H**) Same as (**E**), (**F**), for the scale parameter of the perceptual contrast sensitivity function.

DOI: https://doi.org/10.7554/eLife.43533.015

The following figure supplement is available for figure 6:

**Figure supplement 1.** Bilateral ongoing motif does not affect perceptual reports.
DOI: https://doi.org/10.7554/eLife.43533.016

To estimate the visual component $V(c_i, t)$, we used the average across trials of the activity measured at each stimulus contrast. To estimate the amount of bilateral fluctuation $F_i(t)$ on trial $i$, we measured the activity $f_i'(t)$ contralateral to the predicted area, subtracting any trial-triggered activity $V'(c_i, t)$ that may be seen there:

$$B_i(t) = h\left[f_i'(t) - V'(c_i, t)\right] \tag{2}$$

The weight $h \leq 1$ is a fixed coefficient, and was obtained from trials with no stimulus (0% contrast) by linear regression against the contralateral fluorescence.

The additive model provided good estimates of trial-by-trial activity (**Figure 7B**). Consider again the activity measured in right area PM in response to 100% contrast stimuli presented in the left visual field (as in **Figure 5A**). The bilateral ongoing activity $F_i(t)$ estimated from left area PM showed strong fluctuations across time and trials (**Figure 7B₁**). These fluctuations were often as large as the visually evoked response $V(100, t)$, obtained by averaging activity across trials in right area PM (**Figure 7B₂**). The sum of these two components yields predicted single-trial activities in right area PM (**Figure 7B₃**). This predicted activity closely resembles the observed data (**Figure 7B₄**).

The additive model was most successful when bilateral ongoing fluctuations were measured across symmetrical cortical locations (**Figure 7C**). We asked whether the model would degrade if bilateral activity $F_i(t)$ was measured not from the contralateral PM but rather from other contralateral areas. Doing so strongly decreased the explained variance: explained variance was highest when the region used to estimate the bilateral activity was contralateral PM or regions near it, and decreased with distance from contralateral PM (**Figure 7D**). Similar results were obtained in the other visual areas (**Figure 7—figure supplement 1**), confirming that the bilateral ongoing activity contributing to the observed variability is not global, but rather local to individual areas.

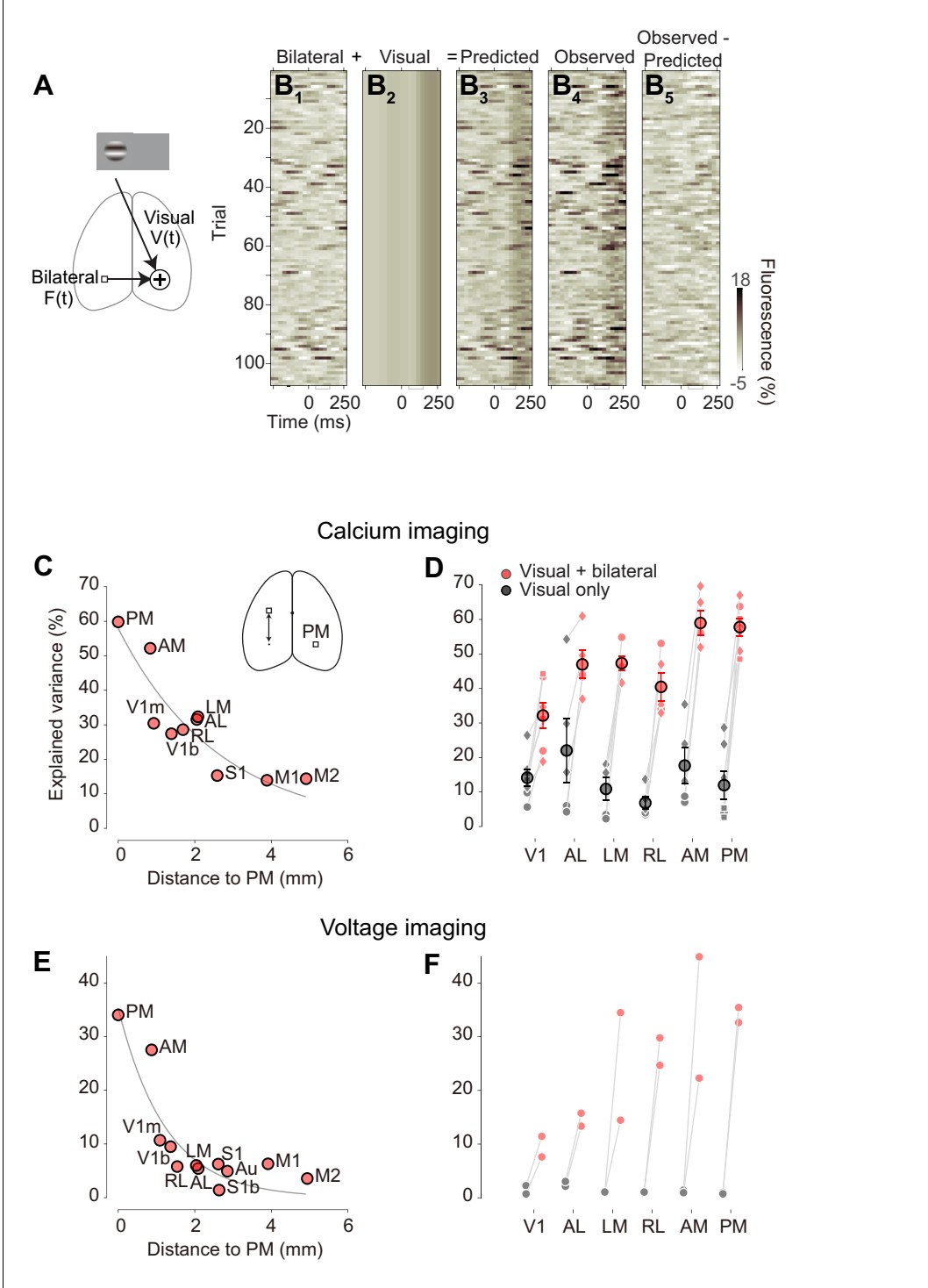

**Figure 7.** Additive model accounts most trial-by-trial variations in visual responses. (**A**) Schematic representation of the additive model (*Equation 1*). (**B**) Single-trial PM traces predicted by the additive model. (**B₁**) bilateral activity, estimated from left PM. (**B₂**) visually evoked response of the right PM, estimated as average across trials of right PM. (**B₃**) predicted activity of the right PM by the additive model. (**B₄**) observed activity of the right PM. (**B₅**) difference between predicted and observed activity in right PM. (**C**) Variance of activity in PM explained by the additive model, as a function of distance from location of PM to location of ROI used as a regressor. Circles represent average across animals. Gray curve indicates a result of exponential fit. (**D**) Variance explained by the visually evoked and bilateral activities (additive model, *red*), and by the visually evoked activity alone (*black*). Circles and error bars indicate average and S.E. across animals. Dots represent different animals. (**E**) As in (**C**), for

*Figure 7 continued on next page*

*Figure 7 continued*
voltage signals (n = 2 mice). Circles represent average across animals. (**F** aAs in **D**) for voltage signals (*circles*, n = 2 mice).
DOI: https://doi.org/10.7554/eLife.43533.017
The following figure supplement is available for figure 7:

**Figure supplement 1.** Variance explained by the additive model in different visual areas.
DOI: https://doi.org/10.7554/eLife.43533.018

This simple additive model worked well in all visual areas, and revealed that bilateral ongoing fluctuations had a much larger effect on cortical activity than evoked visual responses (*Figure 7D*). In area PM, the additive model explained 58 ± 3% of the variance (S.E. n = 7 mice). This is a very high performance, and the bilateral component is crucial to obtain it. Indeed, the visually evoked responses $V(c_i, t)$ were only able to account for 12 ± 4% of the variance. Similar effects were seen in area AM, and to a large extent also in the remaining visual areas (*Figure 7D*).

Moreover, the model performed well also on the data obtained with voltage imaging (*Figure 7E, F*). For area PM, for instance, the additive model accounted for up to 34% of trial-by-trial variability in the voltage signals (*Figure 7E*). This value was significantly higher than the variance explained by the visually evoked responses, which is known to be particularly low for voltage signals (*Figure 7F*). Much of the large variability observed across trials when performing widefield imaging experiments (*Arieli et al., 1996*; *Carandini et al., 2015*), therefore, is not due to imaging noise, but rather to structured ongoing neural activity tightly coordinated across the hemispheres.

## Discussion

By taking advantage of widefield imaging, we monitored both cortical hemispheres in mice performing a visual detection task, and found that bilateral ongoing activity has a powerful additive impact on sensory processing. As reported previously (*Mohajerani et al., 2010*; *Shimaoka et al., 2017*; *Vanni et al., 2017*; *Vanni and Murphy, 2014*), voltage and calcium signals exhibited strong bilateral fluctuations in ongoing activity. In both types of signal, the impact of these bilateral fluctuations varied across cortical areas, and was stronger in areas closer to the midline. Two exceptions were the monocular region of area V1, where activity was less bilateral in all conditions, and area M2, where activity was less bilateral during task engagement. Bilateral ongoing fluctuations continued following the onset of the visual stimulus, and typically swamped the resulting unilateral visual responses. Higher bilateral ongoing activity increased visual responses equally at all contrasts, regardless of response amplitude, indicating that the interaction between ongoing activity and visually evoked responses was additive. A simple additive model provided good estimates of trial-by-trial visual response in all visual areas, and revealed that bilateral ongoing fluctuations had a much larger effect on cortical activity than evoked visual responses.

Bilateral fluctuations have been observed in multiple circumstances, but in the absence of overt movements their functional significance is not clear. The cortex exhibits clear bilateral activity during motor behavior (*Musall et al., 2018*), which is understandable given the symmetry involved in many body movements. It is less clear, however, why there should be bilateral symmetry in sensory responses, and even less clear why there should be symmetry in ongoing activity, both in wakefulness and in anesthesia (*Mohajerani et al., 2010*; *Shimaoka et al., 2017*; *Vanni and Murphy, 2014*).

Regardless of their origin, it is likely that bilateral fluctuations are at least partially mediated by callosal connections. A number of brain structures connect the two cerebral hemispheres, including anterior commissure, cingulum, interhemispheric thalamic connections, and corpus callosum. Among these, the corpus callosum is the largest white matter tract connecting the cerebral hemispheres in placental mammalian brains. Ongoing activity is less correlated across hemispheres in mutant mice that lack the corpus callosum (*Mohajerani et al., 2010*) and in rats after surgical section of the corpus callosum (*Magnuson et al., 2014*). However, ongoing bilateral activity is also reported in humans born without a corpus callosum (*Tyszka et al., 2011*), suggesting that bilateral fluctuations can also be sustained by other brain structures.

Bilateral fluctuation represents shared activity across neurons, which impacts on response variability. Our widefield imaging result conforms to the previous electrophysiological report by

*Schölvinck et al. (2015)*, that ongoing activity is shared across large proportions of neurons and interacts additively with visually evoked responses, and that it accounts for a large fraction of trial-by-trial fluctuations in observed visually evoked activities. The present study extends this idea by demonstrating that such global fluctuation is actually shared across hemispheres, yet in an area-specific manner.

Our results indicate that a full understanding of activity in a population of neurons in one hemisphere may require recordings from the corresponding area of the other hemisphere. Because ongoing activity changed rapidly and was largely bilateral, its impact could not be estimated through the typical procedure of measuring unilateral activity before stimulus onset (*Arieli et al., 1996*). Indeed, compared to bilateral activity, activity in the prestimulus period had a smaller impact on evoked responses than simultaneously measured activity in the contralateral hemisphere. Given the specificity that we have observed in bilateral fluctuations, the ideal placement of a sensor in the contralateral hemisphere would be one that monitored the symmetrical populations to the ones of interest. Experiments with this design are emerging in electrophysiology (*Cohen and Maunsell, 2009*; *Rabinowitz et al., 2015*), and a promising strategy would be to pair them with simultaneous widefield imaging.

Even though bilateral fluctuations occurred in regions necessary for this visual task (*Burgess et al., 2017*), they did not impact choice behavior. Perhaps, a downstream readout mechanism removes the bilateral fluctuations by taking the difference of activity on left and right visual cortices (*Zatka-Haas et al., 2019*). It might thus be easy for the brain to ignore them during perceptual decisions based on vision, even though the fluctuations continue during visual stimulation and have a powerful additive impact on visual processing. One possible test of this hypothesis might be to perform optogenetic stimulation (e.g. *O'Connor et al., 2013*; *Sreenivasan et al., 2016*) both unilaterally and bilaterally, and ask whether the animal finds it easier to ignore bilateral activations during behavior.

## Materials and methods

Experimental procedures were conducted according to the UK Animals Scientific Procedures Act (1986), under personal and project licenses released by the Home Office following appropriate ethics review.

### Transgenic lines

To obtain mice-expressing GCaMP6s in selected neuronal populations, we used Snap25-GCaMP6s (B6.Cg-Snap25tm3.1Hze/J), Jax025111. This mouse line expresses GCaMP6s in all neurons. Alternatively, we crossed two lines of mice: (1) TetO-GCaMP6s Jax024742 (*Wekselblatt et al., 2016*); (2) a line expressing tTA in excitatory neurons, Camk2a-tTA line, Jax007004. The results of this crossing were mice-expressing GCaMP6s in all excitatory neurons.

To obtain mice expressing GCaMP6f, we crossed two lines of mice: (1) the Cre-dependent GCaMP6f reporter line Ai95, Jax028865; (2) a line expressing Cre in Vglut1-expressing cells, Vglut1-Cre, Jax023527. The results of this crossing were mice expressing GCaMP6f in Vglut1-expressing excitatory neurons.

To obtain mice-expressing voltage-sensitive fluorescent protein (VSFP) Butterfly 1.2 in selected neuronal populations, we crossed three lines of mice: (1) the Cre/tTA-dependent VSFP reporter line Ai78, Jax023528 (*Madisen et al., 2015*); (2) a line expressing tTA in excitatory neurons, Camk2a-tTA line, Jax007004; (3) a line expressing Cre in neurons of all layers, Emx1-Cre, Jax005628. The results of this crossing were mice expressing VSFP Butterfly 1.2 of all layers (Emx1-Cre;Camk2a-tTA;Ai78, n = 2).

### Surgery

The nine mice (six males) were implanted with a head post and a thinned or intact skull cranial window (*Drew et al., 2010*) over the dorsal part of the both cortical hemispheres. An analgesic (Rimadyl) was administered on the day of the surgery (0.05 ml, s.c.), and on subsequent 2 days (in the diet). Anesthesia was obtained with isoflurane at 2–3% and kept 1–1.5% during surgery. To prevent dehydration during surgery, saline was administered every hour (0.01 ml/g/h, i.p.). Body temperature was maintained at 37°C using a feedback-controlled heating pad, and the eyes were

protected with ophthalmic gel. The head was shaved and disinfected with iodine, the cranium was exposed, and the bone was thinned for VSFP animals with a dental drill and a scalpel over the dorsal part of the cortex. A 3D-printed light-isolation cone surrounding the frontal and parietal bones was attached to the skull with cyanoacrylate. Subsequently, a metal or 3D-printed head plate was secured with dental cement. After the cement solidified, this opening was filled with transparent cement.

## Behavioral task

The general framework of the task and training procedure is found in *Burgess et al. (2017)*. The sequence of trial events was as follows. First, the mouse kept the wheel still (quiescent period) to initiate the trial. Second, a visual grating stimulus was presented, and, during an 'open loop' period, wheel movements were ignored. Third, a go tone was played, after which point the wheel turns resulted in movements of the visual stimuli ('closed loop'). If the mouse turned the wheel such that the stimulus reached the center of the screen, the mouse received water reward (1–3 ml). If instead the mouse moved the stimulus by the same distance in the opposite direction, this incorrect decision was penalized with a timeout (typically, 2 s) signaled by auditory noise. In either case, the grating remained locked in its response position for 1 s and then disappeared. In this paper, we added slight modifications to the two-alternative task for GCaMP and VSFP animals. For GCaMP animals, we extended the two-alternative tasks by adding a 'no-go' response option when there was no stimulus. The result is the two-alternative unforced-choice (2AUC) task. Training started on the two-alternative forced-choice task, then we constrained the response window to 1.5 s and added the no-go condition: when the stimulus was absent (zero contrast), mice earned the reward by not turning the wheel (no-go; *Figure 3A*) for 1.5 s (*Figure 3B*). For VSFP animals, we used the 2AFC version of the task, where the mice were trained to hold the wheel after the stimulus presentation at least for 800 ms. If the mice did not respond within 5 s, the trial was aborted. When the stimulus was absent (zero contrast), water reward was randomly assigned to either to the left or right wheel turning. We analyzed VSFP activity in all sessions (25 sessions in two mice) and GCaMP activity in 37/61 sessions in seven mice, selected to have >100 trials and overall hit rate >65%.

In subset of mice, we also acquired widefield data during a passive condition, in which the same visual stimuli were applied but the waterspout was retracted. In this condition, the mouse quickly learned to stay still without moving the wheel.

For investigating visually evoked neural activity and ongoing activity without mice's body movement, we ran custom-made algorithm to detect movement of the wheel, eye blinks and saccades (*Engbert and Kliegl, 2003*). Trials including more than one of these events around the time of the visual stimulus onset (GCaMP: −500 ~ 200 ms; VSFP: −500 ~ 800 ms) were excluded from the further analysis.

## Imaging

We monitored GCaMP/VSFP signals with macroscope based on the tandem lens design and epi-illumination system (*Ratzlaff and Grinvald, 1991*), which we used in previous reports (*Carandini et al., 2015*; *Madisen et al., 2015*).

For GCaMP imaging, excitation light was provided by alternating blue (470 nm) and purple (405 nm) LEDs (Cairn OptoLED, P1110/002/000), through an additional band-pass filter (Semrock FF01-466/40-25 for blue, Chroma ET405/20x for purple). Illumination light passed through a dichroic (425 nm; Chroma T425lpxr), and 3-mm-core optical fiber (Cairn Research, P135/015/003), then reflected off another dichroic (495 nm; Semrock FF495-Di03−50 × 70) to the brain. Emitted light was passed through the second dichroic and emission filter (Edmund Optics 525/50–55 (86-963)) to single sCMOS camera (PCO pco.edge 5.5). The light intensity measured at cortical surface ranged from 0.04 to 0.09 mW/mm$^2$ for the blue LED, and from 0.08 to 0.12 mW/mm$^2$ for the purple LED. Imaging was conducted at 35 Hz in each purple and blue channel (70 Hz in total) with a nominal spatial resolution of 21.7 µm/pixel.

For VSFP imaging, excitation light was provided by a blue LED (Brainvision LEX2-B), through a band-pass filter (482 nm; Semrock FF01-482/35) and a dichroic mirror (Semrock FF506-Di03). This blue LED was alternated with red (631 nm) and green (528 nm) LEDs for simultaneous reflectance imaging to estimate blood volume and oxygenation. Fluorescence signals of VSFP FRET

chromophores (mKate2 and mCitrine), as well as the reflectance signals were imaged via two of the sCMOS cameras. When the blue LED was turned on, first camera recorded the emitted fluorescence from mCitrine, which was reflected by a second dichroic mirror (Semrock FF593-Di03), passed through an emission filter (Semrock FF01-543/50-25). The second camera recorded the emitted fluorescence from mKate2, passed through the second dichroic mirror and an emission filter (Semrock BLP01-594R-25). When the red and green LEDs were turned on at one time, the first and the second cameras recorded reflectance of cortical surface emitted by the red and green LEDs, respectively. Imaging was conducted at 35 Hz in each blue and red/green channel (total 70 Hz), with a nominal spatial resolution of 21.7 μm/pixel.

## Signal processing

We used a data compression method by computing the singular value decomposition of the 3D image stack, and storing the top 500 singular values, that is 500 images and 500 time courses (*Steinmetz et al., 2017*). Using 500 components provided excellent fidelity in reconstructing the individual pixel traces (*Figure 1—figure supplement 1*).

For GCaMP data, the acquired fluorescence signal was contaminated by signals originating outside of the target fluorophores expressed in neurons. To reduce this artifact, simultaneously measured fluorescence signal excited by the purple LED was scaled by the regression coefficient with the signal excited by the blue LED, and subtracted out from it. Subsequently, to extract the time of firing from the slow temporal kinetics of GCaMP6s, we applied a temporal derivative and rectification (*Markowitz et al., 2018*). Finally, the signal was lowpass filtered with a cutoff of 7 Hz to attenuate signals from heartbeats. Filtering did not affect the ratio of unilateral to bilateral responses (*Figure 1—figure supplement 1*).

For VSFP data, the red and yellow fluorescence signals were analyzed using the gain-equalization method (*Akemann et al., 2012*; *Carandini et al., 2015*), by equalizing the gains at the heart beat frequency between the two cameras. The gain equalization factors were obtained once per recording session at each pixel-basis. To further exclude the hemodynamic signals in the ratiometric signal, simultaneously measured red and green reflectance signals were scaled by the regression coefficient with the ratiometric signal, and subtracted out from the ratiometric signal (*Ma et al., 2016*). Finally the signal was filtered below 0.5 Hz and above 7 Hz to eliminate signals from heartbeats.

## Identification of sensory areas

Visual stimuli were presented via LCD monitors (Iiyama ProLite E1980) placed 19 cm away from the animal. To identify areas six visual areas (V1, LM, AL, RL, AM, PM), we used vertical or horizontal bars sweeping across the visual field (*Garrett et al., 2014*; *Kalatsky and Stryker, 2003*). We built maps of retinotopy then parcellated the activated region into distinct visual areas by identifying changes in the sign of the retinotopic mapping between the screen and the cortical surface. This sign is positive if clockwise circles in the visual field map to clockwise circles on cortex and negative if they map to anti-clockwise circles (*Sereno et al., 1994*). Additionally in VSFP animals, train of auditory or somatosensory stimuli was used to identify these sensory areas. Somatosensory stimuli were air puffs delivered from a pressure injector (Toohey Company Pressure system IIe) towards the bulk of the whiskers at a pressure of 40 PSI via a silicone tube (0.5 mm open tip diameter). Auditory stimuli were 13 kHz tones at 80 dB SPL via a magnetic speaker (Tucker-Davis Technologies) placed 19 cm away from the animal. We obtained location of the S1 barrel cortex and auditory cortex, by calculating the response amplitude at the stimulation frequency.

## Fitting contrast-response curves

To quantify the contrast sensitivity of the recorded signal, we fitted the contrast-response relationship with the Naka-Rushton function:

$$R(c) = R_0 + R_{max}\frac{C^n}{C_{50}^n + C^n} \tag{3}$$

where $R$ is the neuronal activity, $R_0$ is the background discharge, $R_{max}$ is maximal response, $C$ is the contrast of grating stimuli, $n$ is exponent of the power function, and $C_{50}$ is contrast for half of $R_{max}$.

In each area, we first estimated all the parameters for the curve of grand average. Subsequently, we estimated $R_0$ and $R_{max}$ for individual curves subdivided by ongoing activity.

As noted in the paper, in many occasions we fit these curves to two halves of the data, segregated according to some criterion. We verified that the results remain the same if we change the criterion.

## Fitting single-trial responses

We fitted single-trial signals with the additive model $f_i(t)$ (*Equation 1*). To evaluate how well the model accounts for variability in observed activity $r_i(t)$, we averaged both model and responses over an interval of 50-150 ms after stimulus onset, and we computed explained variance as:

$$100 \left( 1 - \frac{\sum_i (r_i - f_i)^2}{\sum_i (r_i - \bar{r})^2} \right), \tag{4}$$

where $\bar{r}$ is the average response across trials $i$. We averaged this quantity across trials of >0% contrasts to obtain the explained variance in each ROI. Note that the explained variance was measured with cross-validation: weights were estimated based on a training set (0% contrast trials) and used to explain the variance of the test set (>0% contrast trials).

## Fitting psychometric curves

To quantify the contrast sensitivity of the animal's perceptual report, we fitted the choice probability with multinomial logistic regression (similar to *Burgess et al., 2017*). In the model, choices depend on two decision variables, one for choosing L and the one choosing R, depending on the contrast $c_L$ and $c_R$:

$$z_L = b + sg(c_L), \, z_R = b + sg(c_R) \tag{5}$$

Here, $g()$ is the function in *Equation 3*, $b$ represent offset toward choosing left or right relative to no-go, and $s$ measure the weight assigned to visual evidence. The decision variables, in turn, determine the probabilities $p_L$, $p_R$, $p_0$ of choosing L, R, or no-go, and specifically the log odds of choosing correct versus choosing no-go:

$$\log(p_L/p_0) = z_L, \, \log(p_R/p_0) = z_R \tag{6}$$

As the resulting psychometric curves are symmetrical between choosing left and right, we only showed one side in *Figure 6*. We fit the two parameters of the decision variables (*Equation 5*) to the data obtained using all trials per animal or all animals through multinomial logistic regression, and optimized the two additional two parameters describing contrast sensitivity (*Equation 3*). After fixing the parameters describing contrast sensitivity (*Equation 3*), we fit the two parameters of the decision variables (*Equation 5*) again in each of the two trial groups of high and low bilateral ongoing activity.

## Acknowledgements

We thank Miles Wells for mouse training and Charu Reddy for surgical procedure on GCaMP mice. This work was supported by grants from JSPS (to DS), from the Human Frontier Sciences Program (grant LT001071/2015 L to NAS), from the Marie Skłodowska-Curie program (grant 656528 to NAS), and from Wellcome Trust (grant 108726 to KDH and MC). MC holds the GlaxoSmithKline/Fight for Sight Chair in Visual Neuroscience.

## Additional information

### Funding

| Funder | Grant reference number | Author |
|---|---|---|
| Japan Society for the Promotion of Science | | Daisuke Shimaoka |

| Human Frontier Science Program | LT001071/2015-L | Nicholas A Steinmetz |
| H2020 Marie Skłodowska-Curie Actions | 656528 | Nicholas A Steinmetz |
| Wellcome | 108726 | Kenneth D Harris<br>Matteo Carandini |
| Simons Foundation | SCGB 325512 | Kenneth D Harris |

The funders had no role in study design, data collection and interpretation, or the decision to submit the work for publication.

### Author contributions
Daisuke Shimaoka, Conceptualization, Data curation, Software, Formal analysis, Funding acquisition, Investigation, Visualization, Methodology, Writing—original draft, Writing—review and editing; Nicholas A Steinmetz, Resources, Software, Investigation, Writing—review and editing; Kenneth D Harris, Conceptualization, Supervision, Funding acquisition, Writing—review and editing; Matteo Carandini, Conceptualization, Supervision, Funding acquisition, Investigation, Project administration, Writing—review and editing

### Author ORCIDs
Daisuke Shimaoka  https://orcid.org/0000-0002-9435-5448
Nicholas A Steinmetz  http://orcid.org/0000-0001-7029-2908
Kenneth D Harris  http://orcid.org/0000-0002-5930-6456
Matteo Carandini  https://orcid.org/0000-0003-4880-7682

### Ethics
Animal experimentation: Experimental procedures were conducted according to the UK Animals Scientific Procedures Act (1986), under personal and project (70/8021) licenses released by the Home Office following appropriate ethics review.

### Decision letter and Author response
Decision letter https://doi.org/10.7554/eLife.43533.026
Author response https://doi.org/10.7554/eLife.43533.027

## Additional files

### Supplementary files
• Transparent reporting form
DOI: https://doi.org/10.7554/eLife.43533.019

### Data availability
Data has been deposited to Dryad Digital Repository under the doi:10.5061/dryad.rd00gk7.

The following dataset was generated:

| Author(s) | Year | Dataset title | Dataset URL | Database and Identifier |
| --- | --- | --- | --- | --- |
| Daisuke Shimaoka, Nicholas A Steinmetz, Kenneth D Harris, Matteo Carandini | 2019 | Data from: The impact of bilateral ongoing activity on evoked responses in mouse cortex | https://doi.org/10.5061/dryad.rd00gk7 | Dryad Digital Repository, 10.5061/dryad.rd00gk7 |

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
