## [Decision Letter]

Thank you for submitting your article "The impact of bilateral ongoing activity on evoked responses in mouse cortex" for consideration by *eLife*. Your article has been reviewed by two peer reviewers, and the evaluation has been overseen by a Reviewing Editor and Joshua Gold as the Senior Editor. The reviewers have opted to remain anonymous.

The reviewers have discussed the reviews with one another and the Reviewing Editor has drafted this decision to help you prepare a revised submission.

Summary:

Spontaneous neural activity is ubiquitous, but its impact on perception and evoked neural activity is not well understood. The manuscript investigates bilaterally symmetrical cortical activation observed in mice during an awake resting state condition and during a visual discrimination task. The authors extend previous work by a number of labs defining bilaterally symmetrical activity dynamics in the mouse brain. Strikingly, these bilateral dynamics are preserved under conditions of spontaneous activity or even anesthesia states. The authors examine the interactions of this ongoing activity with task-evoked activity. They find that although ongoing activity is of greater amplitude than task-dependent signals, it does not interfere with the ability to perform the task. In addition, other significant findings are reported, such as the lack of a relationship between task-dependent and ongoing activity, and a linear interaction between the spontaneous activity and the evoked activity, with no impact on behavioral performance. The results are very surprising, as most studies have suggested that the ongoing activity will, at least to some degree, affect behavior and evoked responses (e.g. Mcginley et al., Neuron 2015). Mesoscopic imaging of these spontaneous neural activity patterns and its impact on a behavior is an important and novel advance for the field. The paper is well-written and it is relatively clear in its presentation and accessible to a wide range of readers.

Essential revisions:

1) Mesoscopic calcium/voltage imaging is becoming a very important tool in systems neuroscience, but it would be good to show some more details on the data processing pipeline, given the importance and impact of the results shown in this manuscript.

The pre-processing step in the data analysis presented in the paper needs to be more clearly described and discussed. There are concerns that the pre-processing, combined with subsequent band-pass filtering could amplify relatively small parts of the original signal. Could this artificially hide some non-linear interaction between the spontaneous and evoked activity? Does the linear relationship between the evoked and spontaneous activity persist in the raw data?

To address these concerns, particularly that the noise removal steps might remove most of the signal in the 0.5 – 7.0 Hz range, it is suggested that the authors:

– Show analyses both of the raw and "denoised" data, including some example traces before and after pre-processing and bandpass filtering, as well as the power spectra of these signals.

– Perform their denoising SVD in the space-frequency domain (rather than in the temporal domain) (see Prechtl et al., 1997 PNAS), and use the resulting components with substantial power in the 0.5-7 Hz band.

2) The manuscript needs to better define the behavioral state of the animal. Are there particular bilateral limb, whisker, or facial movements during sensory-motor bilateral activity? Videos of the animal body, if available, might help make this point a bit more clear to the reader.

3) The term "ongoing activity" should be more clearly defined.

4) Points made in Figure 3A such as ongoing activity dominating stimulus-dependent activity should be stated quantitatively. The current draft only states when "acquiring a large number of trials could ongoing fluctuations be averaged out". Please state what a "large number of trials" is and exactly what the impact can be. Presumably, this effect is dependent on where ongoing spontaneous activity occurs and what particular spatial temporal characteristics it contains. One could imagine ongoing activity could have discreet motifs, as described in Mohajerani et al., 2013, which should also be referenced.

5) The authors elegantly show that bilateral activity adds linearly to stimulus trial evoked activity. While the authors have clearly defined this, it needs to also be stated that this conclusion only holds for a visual stimulus. In the experiments, sequences of activity were chosen where mice were attending to visual stimuli and not doing other tasks such a forelimb or whisker-dependent task. It is possible that most of the ongoing activity has origins outside the visual system, making it unlikely that ongoing activity would interfere with visual processing. In contrast, if the task were a whisker or forelimb-dependent one, and the mouse was also moving these body parts, very different results could be obtained. Please mention this caveat in the Abstract and also discuss it more in the manuscript.

5b) The authors should contrast their findings on ongoing visual-like activity and task success rate and response additivity with activity that features different sensory systems (perhaps forelimb and whisker). One way to do this is by using a template-matching or similar scheme as in, e.g. Mohajerani et al., 2013, to find the visual-system-like versus whisker-like ongoing activity and see whether these have any specific impact on a visual task. It would be good to see a quantification of the general composition of the ongoing activity: how much is whisker- versus visual-like, etc. These would be additional analyses, but require no new data collection.

To add to the Discussion section:

6) In other recent work, it has been shown that a form of ongoing activity derived from optogenetic stimulation can bias behavioral activity. Why does optogenetic stimulation have an impact, but not the forms of spontaneous ongoing activity monitored here? e.g. O'Connor et al., 2014 and Sreenivasan et al., 2016

7) Can one conclude that the ipsilateral hemisphere really is independent during the visual task? Perhaps, instead, it gets inhibited, potentially dampening its utility for reporting the impact of ongoing activity on evoked activity?

---

## [Author Response]

Essential revisions:1) Mesoscopic calcium/voltage imaging is becoming a very important tool in systems neuroscience, but it would be good to show some more details on the data processing pipeline, given the importance and impact of the results shown in this manuscript.The pre-processing step in the data analysis presented in the paper needs to be more clearly described and discussed. There are concerns that the pre-processing, combined with subsequent band-pass filtering could amplify relatively small parts of the original signal. Could this artificially hide some non-linear interaction between the spontaneous and evoked activity? Does the linear relationship between the evoked and spontaneous activity persist in the raw data?To address these concerns, particularly that the noise removal steps might remove most of the signal in the 0.5 – 7.0 Hz range, it is suggested that the authors:– Show analyses both of the raw and "denoised" data, including some example traces before and after pre-processing and bandpass filtering, as well as the power spectra of these signals.– Perform their denoising SVD in the space-frequency domain (rather than in the temporal domain) (see Prechtl et al., 1997 PNAS), and use the resulting components with substantial power in the 0.5-7 Hz band.

We thank the reviewers for these comments, which indicate that our explanation was not clear. We have now edited the Materials and methods to explain that we used SVD only to compress the data and speed up the analysis, not for denoising. Indeed, as we now explain, the 500 components that we use capture close to 100% of the variance (Author response image 1). Had we wanted to denoise, we could have used a much smaller number of components (e.g. 20-50) with minimal loss. So, the compression we applied is essentially lossless, and its denoising effect is minimal.

**Author response image 1. respfig1:** Singular Value Decomposition efficiently compresses calcium-imaging data. A few tens of components are sufficient to capture close to 100% of the variance, both in the calcium-sensitive channel (blue) and in the calcium-independent channel (red). In the paper, we use 500 components (arrow). Shaded area represents s.d. across imaging sessions. Similar results were obtained with Voltage-Sensitive Fluorescent Protein imaging.

To illustrate this point, we plotted an example trace reconstructed from different numbers of components, from 50 to 2,000 (Author response image 2). The traces are essentially identical. This confirms that the 500 components we used were more than sufficient.

**Author response image 2. respfig2:** Example GCaMP traces in V1, using 5 numbers of SVD components, from 50 to 2,000 components. The resulting five traces look identical. The number of components used in the paper is 500.

Regarding the filtering operation (the derivative and the subsequent lowpass filtering), we checked whether it affects the results of the paper, and it does not. As illustrated in Author response image 3, the filtering has a strong effect on the power spectral density (as expected), but it does not affect the relative strength of bilateral vs. unilateral activity, which is independent of frequency.

**Author response image 3. respfig3:** Bandpass filtering the signals does not affect the estimate of bilateral activity. Left: Results obtained after taking the derivative and lowpass filtering (as used in our paper). Right: Results obtained without any filtering. By definition, filtering affects the power spectral density (top). However, the relative strength of bilateral activity (relative to total) remains the same, independent of frequency (bottom). We now show the bottom left panel in Figure 1.

In making Author response image 3, we realized that the ratio of power spectral density (shown in the bottom panels) is a better graph than the one we had in our Figure 1. It is better because it illustrates directly the result (i.e. that bilateral activity is independent of frequency), and is independent of whether we filter the data or not. Therefore, we updated Figure 1D,H in the paper, to show this quantity.

2) The manuscript needs to better define the behavioral state of the animal. Are there particular bilateral limb, whisker, or facial movements during sensory-motor bilateral activity? Videos of the animal body, if available, might help make this point a bit more clear to the reader.

We completely agree, and we have now added two videos: one for the three example trials in Figure 1 (where the visual stimulus was absent, i.e. 0 contrast) and the other for the three example trials in Figure 3 (where the visual stimulus was present). The videos show not only the cortical activity but also the snout and the eye of the mouse, and traces for wheel movements and licks. These videos confirm that indeed in the intervals that we analyzed (up to 200 ms after stimulus onset) there is no overt movement.

It is possible, however, that with videos taken from other angles and with better lighting, we would have seen movement. In future studies, we hope to place more cameras around the mouse, although this is made difficult by the presence of the three screens.

3) The term "ongoing activity" should be more clearly defined.

We have now edited Abstract and Introduction to define “ongoing activity” as the patterns of activity exhibited by the awake cerebral cortex even in the absence of external stimuli or overt behavior.

4) Points made in Figure 3A such as ongoing activity dominating stimulus-dependent activity should be stated quantitatively. The current draft only states when "acquiring a large number of trials could ongoing fluctuations be averaged out". Please state what a "large number of trials" is…

We agree. We now specify in Figure 3 and in the text that the analysis included 258 trials for voltage imaging and 104 trials for calcium imaging.

…and exactly what the impact can be. Presumably, this effect is dependent on where ongoing spontaneous activity occurs and what particular spatial temporal characteristics it contains. One could imagine ongoing activity could have discreet motifs, as described in Mohajerani et al., 2013, which should also be referenced.

We have addressed this comment (and a later one in the same spirit) with a new analysis, shown in two new supplementary figures. In this analysis, we asked whether there were motifs in the ongoing activity that resembled the pattern of activity observed during stimulus responses. We found a continuum of correlations with this pattern of activity (Figure 5—figure supplement 1), and we divided trials in two groups: Those where bilateral activity had positive correlation with this pattern and those where bilateral activity had negative correlation this pattern. The difference in activity between these two conditions was again consistent with an additive interaction (Figure 5—figure supplement 1). Behavior was not significantly different in the two conditions (Figure 6—figure supplement 1). We also added a citation to the paper by Mohajerani and Murphy, 2013, which reported the existence of motifs in unilateral measurements, and to an earlier paper by Kenet et al. in cats.

5) The authors elegantly show that bilateral activity adds linearly to stimulus trial evoked activity. While the authors have clearly defined this, it needs to also be stated that this conclusion only holds for a visual stimulus. In the experiments, sequences of activity were chosen where mice were attending to visual stimuli and not doing other tasks such a forelimb or whisker-dependent task. It is possible that most of the ongoing activity has origins outside the visual system, making it unlikely that ongoing activity would interfere with visual processing. In contrast, if the task were a whisker or forelimb-dependent one, and the mouse was also moving these body parts, very different results could be obtained. Please mention this caveat in the Abstract and also discuss it more in the manuscript.

We see the reviewers’ point: our results are specific to a visual task, and we cannot claim that they would hold for other sensory tasks. To limit our claim we replaced “sensory” with “visual” in Abstract and throughout the last paragraph of Discussion, where we now explain that our conclusions apply to perceptual decisions based on vision.

5b) The authors should contrast their findings on ongoing visual-like activity and task success rate and response additivity with activity that features different sensory systems (perhaps forelimb and whisker). One way to do this is by using a template-matching or similar scheme as in, e.g. Mohajerani et al., 2013, to find the visual-system-like versus whisker-like ongoing activity and see whether these have any specific impact on a visual task. It would be good to see a quantification of the general composition of the ongoing activity: how much is whisker- versus visual-like, etc. These would be additional analyses, but require no new data collection.

As mentioned above, we followed this advice and divided the trials according to whether the bilateral ongoing activity resembled the normal activation of the visual system during stimulus presentation. The results are shown in the new supplementary figure (Figure 6—figure supplement 1). They confirm that there was no impact on behavioral performance: the performance of the animal was similar regardless of whether the bilateral fluctuations correlated positively or negatively with the pattern of activity seen during visual stimulation.

To add to the Discussion section:6) In other recent work, it has been shown that a form of ongoing activity derived from optogenetic stimulation can bias behavioral activity. Why does optogenetic stimulation have an impact, but not the forms of spontaneous ongoing activity monitored here? e.g. O'Connor et al., 2014 and Sreenivasan et al., 2016

We were surprised to find that bilateral activity does not affect visual perception. We do not know if activity elicited by bilateral optogenetic stimulation might similarly be ignored during behavior. We now remark on this in Discussion, as suggested, and we cite the two papers that were suggested (O’Connor et al., 2013, Sreenivasan et al., 2016). However, optogenetic activation in these papers was unilateral. We do not know if bilateral activation would have had different effects.

7) Can one conclude that the ipsilateral hemisphere really is independent during the visual task? Perhaps, instead, it gets inhibited, potentially dampening its utility for reporting the impact of ongoing activity on evoked activity?

In our data we could not detect changes in either the mean or standard deviation of cortical activity ipsilateral to the stimulus, for a single stimulus presented alone. We therefore have no evidence that the ipsilateral side is inhibited, and instead conclude that its activity is independent.